

# Children's physical activity and sedentary time compared using assessments of accelerometry counts and muscle activity level

Ying Gao[1], Martti Melin[1], Karoliina Mäkäräinen[1], Timo Rantalainen[1], Arto J. Pesola[2], Arto Laukkanen[3], Arja Sääkslahti[3] and Taija Finni[1]

[1] Neuromuscular Research Center, Faculty of Sport and Health Sciences, University of Jyväskylä, Jyväskylä, Finland
[2] Active Life Lab, South-Eastern Finland University of Applied Sciences, Mikkeli, Finland
[3] Faculty of Sport and Health Sciences, University of Jyväskylä, Jyväskylä, Finland

Corresponding author
Ying Gao, ying.y.gao@jyu.fi

## ABSTRACT

**Background**. This research compared accelerometry (ACC)-derived and muscle electromyography (EMG)-based estimates of physical activity (PA) and sedentary time in typical PA tasks and during the daily lives of children.

**Methods**. Data was included from two exploratory studies. In Study I, 6–7-year-old children ($n = 11$, 64% girls) were assessed for eight PA tasks (walking, stair negotiation, climbing, crawling, swinging, balancing, trampoline jumping and a game of tag). In Study II, 7–9-year-old children ($n = 14$, 38% girls) were assessed for six PA tasks (walking, sitting, static squat, single leg hops, jump for height and standing long jump), and daily PA during one day with and one day without structured exercise. Quadriceps and hamstring muscle activity and inactivity using EMG shorts and acceleration by waist-mounted accelerometer were simultaneously measured and classified as sedentary, light, moderate and vigorous activity. Data from ACC was further analyzed using five different published cut-off points and varying time windows ($1-60$ s) for comparison with EMG.

**Results**. In the PA tasks ACC counts and EMG amplitude showed marked differences in swinging, trampoline jumping, crawling, static squat, single leg hops, standing long jump and jump for height, the difference being over 170% when signals were normalized to that during walking. Furthermore, in walking, swinging, trampoline jumping, stair negotiation and crawling ACC classified over 60% of the time as vigorous-intensity activity, while EMG indicated primarily light- and moderate-intensity activities. During both days with and without exercise, ACC resulted in greater proportion of light activity ($p < 0.01$) and smaller proportion of moderate activity compared to EMG ($p < 0.05$). The choice of cut-off points and epoch length in ACC analysis influenced the classification of PA level and sedentary time. In the analysis of daily activities the cut-off points by *Evenson et al. (2008)* with epochs of 7.5 s and 15 s yielded the smallest difference (less than 10% of recording time at each intensity) against EMG-derived PA levels.

**Discussion**. This research provides novel insight on muscle activity and thereby on neuromuscular loading of major locomotor muscles during normal daily activities of children. While EMG and ACC provided similar estimates of sedentary time in

13 typical PA tasks, duration of light, moderate and vigorous PA varied considerably between the methods especially during walking, stair negotiation, crawling, swinging and trampoline jumping. *Evenson et al.*'s (*2008*) cut-off points with ≤15 s epoch provided similar classification of PA than EMG during daily life. Compared to impacts recorded using ACC, EMG can provide understanding on children's neuromuscular loading during motor tasks that is useful when studying effects of PA interventions on, and development of, motor competence and coordination.

## INTRODUCTION

A physically active childhood enhances a physically active lifestyle over a life span (*Telama et al., 2014*). It is well documented that physical activity (PA) is associated with numerous physical, psychological/social, and cognitive health benefits and can reduce several risk factors for chronic diseases, e.g., overweight and obesity (*Strong et al., 2005*; *Poitras et al., 2016*). Based on World Health Organization recommendations (2010), children are encouraged to engage in at least 60 min per day of moderate-to-vigorous-intensity physical activity (MVPA) (*WHO, 2010*). However, children spend on average 8 to 9 hours/day in sedentary activities (*LeBlanc et al., 2015*) and sedentary behavior (SB) tends to increase ~30 min per day per year in school-aged children (*Tanaka, Reilly & Huang, 2014*). A rapidly growing body of evidence shows excessive SB to be linked with the accumulation of various health risks, decreased fitness, low self-esteem and decreased academic achievement in school-aged children and youth (5–17 year-old) (*LeBlanc et al., 2015*; *Carson et al., 2016*). It is therefore necessary to accurately classify both SB and PA profiles of children in order to understand their association with health and wellbeing and to inform future intervention programs and PA guidelines.

Accelerometry (ACC) is widely used for the objective monitoring of PA and sedentary time (*Migueles et al., 2017*). Typically, ACC is used to detect accelerations via body movements that are converted to a quantifiable measure such as counts (*Chen & Bassett, 2005*; *Godfrey et al., 2008*) or mean amplitude deviation (*Vähä-Ypyä et al., 2015*), which are validated to estimate metabolic loading of PA. Activities are typically categorized into sedentary, light, moderate and vigorous based on activity counts over epochs of a few seconds to a minute (*Migueles et al., 2017*). The cut-off points for the various categories are based on metabolic equivalents (MET, equals to 3.5 ml/kg/min) derived from oxygen uptake measurements (≥3 METs for moderate, ≥6 METs for vigorous) or observation of tasks or behaviors with specific METs value (e.g., sitting refers to sedentary, walking to moderate and running to vigorous) (*Migueles et al., 2017*). In children, however, the use of standard METs as the reference for cut-off points to classify PA intensity has been questioned (*Harrell et al., 2005*; *Ridley & Olds, 2008*; *Saint-Maurice et al., 2016*). Although several different cut-off points have been reported and used in children's studies (*Migueles*

*et al., 2017*), the established thresholds may vary depending on sample size, age, device version, data processing, reference method and selected tasks, such as in the published methods of *Evenson et al. (2008)*, *Van Cauwenberghe et al. (2011)*, *Pate et al. (2006)*, *Puyau et al. (2002)* and *Sirard et al. (2005)*. Besides the cut-off values, the epoch length (analysis window duration) and band-pass filters (high and low) for sedentary, light, moderate and vigorous activity may also influence the outcomes and their interpretations in children (*Ojiambo et al., 2011*). Thus, methodological considerations are meaningful when assessing PA and sedentary time using ACC.

It is important to note that during childhood not only the cardiorespiratory (metabolic) but neuromuscular system in particular plays an important role as a major contributor to the development of motor performance, fitness, as well as proficiency of gross motor skills (*Haywood & Getchell, 2014*). Because of the nature of accelerometers and the chosen analysis methods, measuring acceleration does not necessarily reflect neuromuscular loading of PA. Relying on the counts based proxy for metabolic cost likely misclassifies quiet standing as SB (*Kozey-Keadle et al., 2011*) although standing increases muscle activity and energy expenditure as compared to sitting (*Mansoubi et al., 2015*; *Gao et al., 2017*). It is important to note that increases in energy expenditure, when people are physically active, are due to the activation of skeletal muscles (*Caspersen, Powell & Christenson, 1985*). When activated, muscle's metabolic rate is increased rapidly (and dramatically, 30–50 times of that during resting *McClave & Snider, 2001*; *Egan & Zierath, 2013*) and overall energy expenditure is increased (*Gao et al., 2017*). On the other hand, lack of muscle contractions (and consequent lack of metabolic stimulus) may be a driver of the aforementioned adverse health outcomes associated with SB, although the possible underlying mechanisms are likely complex and currently unclear (*Hamilton, Hamilton & Zderic, 2007*; *Hamilton, 2017*). Nevertheless, recordings of muscle activity can provide information on the primary stimulus for increased energy expenditure.

Electromyography (EMG) can provide complementary and additional information of the entire PA spectrum by quantifying muscle activity during daily activities (*Tikkanen et al., 2013*; *Tikkanen et al., 2014*). Specially, we have used novel textile EMG shorts for multiple-day recordings of main locomotor muscle activity and inactivity during normal daily life (*Finni et al., 2007*; *Finni et al., 2014*; *Tikkanen et al., 2013*; *Pesola et al., 2015*). We have previously reported in adults that EMG shorts predict energy expenditure across a range of PA intensities particularly well when individualized calibrations are used (*Tikkanen et al., 2014*). Moreover, EMG provides insight into neuromuscular control (*Vigotsky et al., 2018*), which is essential when studying the development of motor skill competence in childhood (*Keawutan et al., 2014*). The advantage of EMG lies not only in its nature to reflect the primary response, e.g., muscle's metabolic loading to PA (*Kemppainen et al., 2002*), but also in the sensitivity of EMG to detect low intensities (*Pesola et al., 2016*) and typically the instantaneous and sporadic activities of children (*Baquet et al., 2007*; *Laukkanen et al., 2013*; *Laukkanen et al., 2014*; *Poitras et al., 2016*). Thus, EMG as a measure of muscle activity reflects both metabolic activity of muscles but also characterizes the neuromuscular function. It is expected that the detailed measurement of muscle activity by using EMG may deepen our understanding of patterns of PA and SB in children.

The purpose of this exploratory research was to compare ACC and muscle EMG activity-derived estimates of PA and sedentary time in 13 typical PA tasks and during the daily lives of children. Due to the nature of ACC to record impacts and EMG to reflect muscle activity, we hypothesized that the mechanical and neuromuscular loading characterized by these different assessment methods categorize several tasks differently in typical PA behaviors of children, and therefore also during daily life. Furthermore, to inform about the selection of cut-off values and epoch lengths for classification of PA intensity levels when using ACC, we compared the outcome of commonly used values to the results obtained using EMG.

## MATERIALS & METHODS

This exploratory research included two independent studies, which both assessed different PA tasks occurring typically during children's daily lives (Study I & II). Combining the studies allowed characterization of physiological requirements with the two different methods, ACC and EMG, from total of 13 different tasks typically occurring in children's daily lives. Study II further measured children's PA on a day with and on a day without structured exercise allowing methodological comparison when estimating daily PA levels.

### Recruitment and study sample

*Study I.* A total of 18 first grade children from one central Finland elementary school volunteered to participate in Study I. Of these participants, 11 ($6.7 \pm 0.5$ years, 63.6% girls, height $127.0 \pm 3.6$ cm, body mass $26.6 \pm 2.5$ kg, BMI $16.5 \pm 1.6$ kg/m$^2$) were included in the final sample. The reason for exclusion ($n = 7$) was insufficient data containing excessive artefacts in one or more EMG channels.

*Study II.* A total of 14 volunteers ($8.6 \pm 0.8$ years, 35.7% girls, height $130.9 \pm 8.0$ cm, body mass $28.1 \pm 4.3$ kg, BMI $16.3 \pm 1.3$ kg/m$^2$) were recruited from sports clubs in central Finland and were measured over two days. Of these participants, ACC data from 14 and EMG from 12 was included when analyzing the typical PA tasks. During daily life successful recordings using ACC were made in all 14 participants on one day and in 13 participants on both days. EMG recordings were successful for one day in 10 participants and for both days in seven participants.

Both Study I (26.8.2014) and Study II (25.8.2012) received ethics approval from the Ethics Committee of the University of Jyväskylä. During recruitment the study purpose, all the procedures, benefits and risks were explained and children were informed that they could decline from any part of the study without any consequences. All children provided oral consent and their legal guardians provided written informed consent before any measurements. The studies were conducted in agreement with the Declaration of Helsinki.

### Study design and protocol

EMG and ACC were measured simultaneously during 13 typical PA tasks on weekdays. Study I assessed walking, stair negotiation, climbing, crawling, swinging, balancing, trampoline jumping and game of tag. These assessments were done during afternoon hours. In Study II, assessment of walking, sitting (on the floor), static squat, single leg hops,

**Table 1** Average (±SD, standard deviation) durations of each task in Study I and Study II. In accelerometry analysis 1 s non-overlapping epochs were used in all tasks.

| Study I | | | Study II | | |
|---|---|---|---|---|---|
| Task | min:sec | SD | Task | min:sec | SD |
| Walking | 00:17 | 00:04 | Walking | 00:31 | 00:11 |
| Stair negotiation | 00:27 | 00:06 | Sitting | 00:31 | 00:15 |
| Climbing | 00:31 | 00:07 | Static squat | 00:21 | 00:11 |
| Crawling | 00:07 | 00:01 | Single leg hops | 00:04 | 00:01 |
| Swinging | 01:13 | 00:05 | Jump for height | 00:01 | 00:00 |
| Balancing | 00:15 | 00:13 | Standing long jump | 00:02 | 00:01 |
| Trampoline jumping | 01:52 | 00:04 | | | |
| Game of tag | 00:51 | 00:08 | | | |

jump for height and standing long jump were done during morning after breakfast. In addition, Study II further monitored the PA level during one day with and one day without structured exercise that took place in a sports club. Data during walking, sitting and static squat from five participants were recorded on both days and analyzed for reliability (Supplemental Information 1 Sheet 6. EMG reliability).

In both studies, the participants were first introduced to ACC and EMG devices. Then they were instructed and helped to put on EMG shorts (Myontec Ltd, Kuopio, Finland) and a waist-mounted accelerometer (X6-1a, Gulf Coast Data Concepts Inc., Waveland, MS, USA) that was attached using an elastic belt. Both devices were set to record throughout the protocol with synchronizing arranged using the same computer clock. Then the participants were asked to perform the PA tasks according to a standard protocol. The instruction given to the participants is included in Supplemental Information 1 (Sheet 5. PA task instructions).

The average duration of each PA task ranged from 1 s to 2 min depending on the type of task (Table 1). The total measurement time was about 30 min and participants were given *ad libitum* rest time between tasks to prevent accumulation of fatigue. All PA tasks were timed and recorded in a log sheet.

### Day with and without structured exercise

In Study II, after completing the PA tasks, the devices were left intact to record activities during the remaining day. These day-long recordings were repeated on schooldays that were selected to be as similar as possible (e.g., neither of days or both days included physical education, same duration of school day) with the exception that one day included structured exercise in a sports club. The exercise training (football or floorball) lasted 60–90 min during the afternoon/evening hours. The participants were asked to remove the recording devices in the evening.

### Recordings and analysis

Full duration of the *PA tasks* (Table 1) was analyzed for both ACC and EMG. In ACC analysis a 1 s epoch was used. In tasks of single leg hops, jump for height and standing long jump, both push-off and landing were included in the analysis.
**Table 2  Cut-off points for different physical activity intensities (15 s) in selected validation studies.**

| Method | Cut-off points: Sedentary | Lightubrk intensity | Moderate- to Vigorous-intensity |
|---|---|---|---|
| *Evenson et al. (2008)* | 12 | 508 | 719 |
| *Pate et al. (2006)* and *Pfeiffer et al. (2009)* | 38 | 420 | 842 |
| *Puyau et al. (2002)*[a] | 200 | 800 | 2,050 |
| *Van Cauwenberghe et al. (2011)* | 373 | 585 | 881 |
| *Sirard et al. (2005)* | 399 | 891 | 1,255 |

Notes.

[a]The original cut-off points derived using 60 s epoch length were divided by 4 to be comparable with 15 s values obtained in other studies.

*For the day-long recordings* analysis was performed on about 9 h period at the same time of the day during both days with and without exercise. In case recordings on one day were shorter than on the other, this procedure eliminated problems in comparing the days but shortened the evaluated time slightly. Successful simultaneous ACC and EMG recordings were required for inclusion into analysis.

*Accelerometry.* A triaxial accelerometer ($\pm 6$ g, 16-bit A/D conversion, sampling at 40 Hz) was secured onto the anterior waistline with an elastic belt at the level of the L4–L5. The resultant vector $\sqrt{x^2 + y^2 + z^2}$ of the triaxial accelerometer signal was calculated, band pass filtered (0.25 Hz to 11 Hz), and a 0.05 g dead-band was digitally applied. Integration and filtering of accelerometer signals were performed in MATLAB software (MathWorks, MA, USA) and converted to values corresponding to ActiGraph GT3X (*Laukkanen et al., 2014*). The conversion factor was obtained from simultaneous recordings with both X6-1a and ActiGraph GT3X devices in children (*Laukkanen et al., 2014*). Data were summed over 15 s epochs with the mean of accelerometer counts less than 12 counts classified as sedentary time, 12–508 counts as light-intensity activity, 508–719 counts as moderate-intensity activity, and more than 719 counts as vigorous-intensity activity (*Evenson et al., 2008*, thresholds for multi-axial accelerometer). To account for epoch length effects, data were also summed over either in 1 s, 7.5 s, 30 s or 60 s using time-window accommodated thresholds from *Evenson et al. (2008)* (e.g., for sedentary threshold 1s is 0.8 counts; 7.5 s is six counts, 15 s is 12 counts, 30 s is 24 counts, 60 s is 48 counts). Furthermore, the data was analyzed using 15 s epochs and cut-off thresholds from *Van Cauwenberghe et al. (2011)*, *Pate et al. (2006)*, *Puyau et al. (2002)* and *Sirard et al. (2005)* (Table 2). These methods were chosen because they are most often used in children's PA research (*Migueles et al., 2017*), but they rely on different criterion measures (e.g., oxygen uptake or observation) and sample (e.g., different age and population). To allow comparison with EMG, ACC counts were normalized to walking (e.g., counts value during walking was considered as 100%).

*Textile EMG shorts* were used to measure muscle activities from the quadriceps and the hamstring muscles. The shorts were made of knitted fabric similar to elastic clothes, into which textile EMG electrodes were embedded in order to measure EMG from the skin surface. Electrodes were positioned in a bipolar configuration over the muscle bellies of the left and right quadriceps (the conductive area of 18 cm$^2$) and hamstring muscles (the

conductive area of 12 cm$^2$). Reference electrodes (the conductive area of 22 cm$^2$) were located longitudinally over tractus iliotibialis. Electrode paste (Redux Electrolyte Crème, Parker, Inc., Fairfield, NJ, USA) was used to minimize the skin-electrode impedance. The device applied an analog bandpass filter (50 Hz to 200 Hz), and sampled at 1000 Hz after which it pre-processed the data into non-overlapping 40 ms root-mean-squared values, which are stored in a small waist-mounted module (*Finni et al., 2007*). These EMG shorts have been tested in adults for validity, repeatability and feasibility, and a detailed description of the recording devices has been reported (*Finni et al., 2007*; *Pesola et al., 2014*). In the present study the day-to-day reliability was assessed in five participants and found to be good (Supplemental Information 1 Sheet 6. EMG reliability). The EMG signals were normalized channel by channel to EMG values measured during normal walking in both Study I and Study II. The normalized EMG from right and left legs were averaged from analysis window to produce the mean quadriceps and hamstring muscle EMG, and further averaged to produce the mean thigh muscle EMG. Classification into different PA categories was based on individually obtained EMG values: (1) EMG amplitude <3 μV was classified as muscle inactivity, (2) amplitudes between 3 μV and the mean EMG value during normal walking as light activity, (3) amplitudes between mean EMG during walking and 2*mean EMG during walking as moderate activity, and (4) amplitudes above 2*mean EMG during walking as vigorous activity (*Tikkanen et al., 2013*; *Pesola et al., 2015*; *Pesola et al., 2016*).

## Statistical analysis

Statistical analyses were conducted using IBM SPSS for Windows 22.0 (SPSS Inc., Chicago, IL, USA). Values are reported as means ± standard deviations unless otherwise indicated. Tests of normality (Shapiro–Wilk) were applied. Repeated measures analysis of variance (ANOVA) was used to compare the main effects of day (exercise vs. no exercise) and device (ACC vs. EMG) and their interaction. ANOVA was also used to examine days with and without exercise separately to examine whether there are main effects of intensity (sedentary, light, moderate, vigorous) and device (ACC vs. EMG), and their interaction. Comparison of EMG and different ACC analysis methods (*Evenson et al., 2008*; *Puyau et al., 2002*; *Sirard et al., 2005*; *Pate et al., 2006*; *Van Cauwenberghe et al., 2011*) were also examined with ANOVA and main effects of the method and intensity and their interaction were examined. Similarly, ANOVA was applied to test significant differences and interaction between epoch length (Evenson's method with 1 s, 7.5 s, 15 s, 30 s, 60 s and EMG) and intensity. Violations of sphericity were corrected using Huynh-Feldt correction and ANOVA was followed by Bonferroni's post hoc for multiple comparisons if necessary to localize the difference. During children's daily lives, agreement between ACC and EMG was evaluated using the Bland-Altman method (*Bland & Altman, 1986*) and for each PA intensity Pearson's correlation coefficients were calculated in order to check if there was heteroscedasticity. A probability level of $p < 0.05$ (two-tailed) was considered statistically significant.

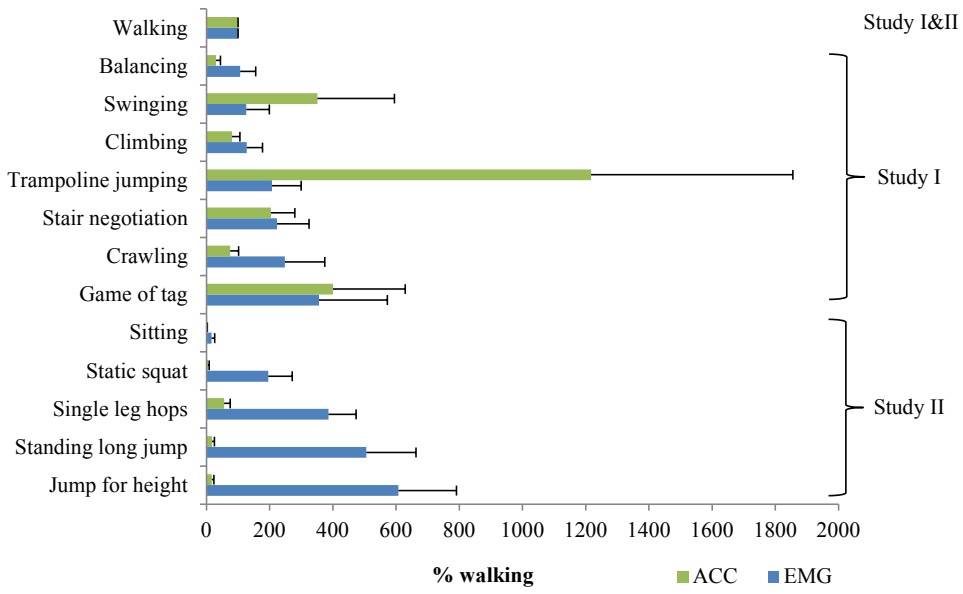

**Figure 1** **Comparison of accelerometry (ACC) counts and electromyography (EMG) amplitude during different PA tasks to children.** Both ACC counts and EMG amplitude are normalized to values obtained during individually chosen normal, preferred walking. All the ACC data in the figure was analyzed using 1 s epoch. Data from both Study I and II were included in walking ($n = 25$), data from Study I: Balancing through Game of tag ($n = 11$), Data from Study II, Sitting through Jump for height ($n = 14$, samples averaged from 2 days). In Study II, EMG data was missing from two participants.

## RESULTS

### ACC counts vs. EMG amplitude in PA tasks

In order to compare ACC counts and EMG amplitude, both were presented relative to normal walking, where ACC corresponded to an average of 1,218 counts (SD 362, range 678–2,033). Figure 1 shows ACC counts and EMG amplitude in various PA tasks when normalized to walking. In swinging, trampoline jumping, crawling, static squat, standing long jump, single leg hops and jump for height, there were marked differences (over 170%) between normalized ACC and EMG values with a range of −590%–1,010%. The mean ACC counts ranged from 18 counts in sitting to 13,566 counts in trampoline jumping. The range of the normalized EMG amplitude was from 16% in sitting to 607% in jumping for height.

### PA levels in typical PA tasks

Figure 2 shows the distribution of PA intensities for each task when assessed using ACC and EMG. EMG-derived muscle inactivity time (range of 0.0%–0.5%) was similar to sedentary time assessed using ACC (range of 0.2%–6.3%). Duration of light-, moderate- and vigorous intensity varied considerably between ACC and EMG in several PA tasks. Walking, stair negotiation, crawling, swinging, trampoline jumping and game of tag were all activities where ACC showed that over 60% of the time was vigorous activity. However, EMG showed that none of the activities reached such high proportion (over 60% of time) of vigorous activity.

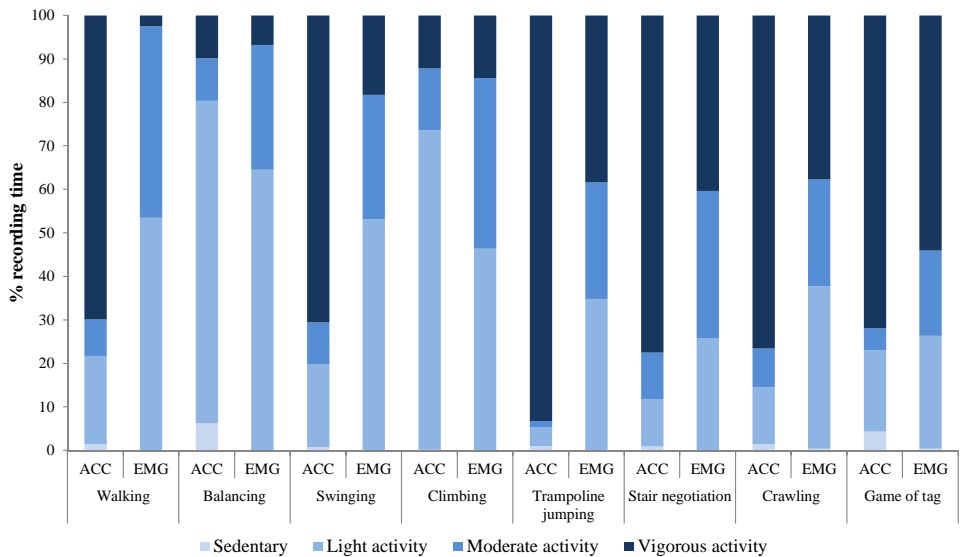

**Figure 2** Proportion of time in sedentary, light, moderate and vigorous activity intensity level in various tasks compared with accelerometry (ACC) and electromyography (EMG). Mean data from Study I, $n = 11$.

## PA levels during daily life

The comparison of PA levels on days with and without exercise is shown in Table 3. ANOVA showed no main effects of day or device, or interaction. When the days were examined separately, both a day with exercise and a day without exercise revealed significant main effects for device ($p = 0.001$ and $p = 0.032$, respectively) and intensity ($p < 0.001$ for both days) and device*intensity interaction ($p = 0.007$ and $p = 0.013$, respectively). Post hoc analysis revealed that time spent in light activity ($p = 0.001$ and $p = 0.002$) and moderate activity ($p = 0.031$ and $p = 0.029$) were significantly different between the methods on days with and without exercise, respectively (Table 3). On average, ACC yielded 9.4% ($p < 0.001$) greater amount of light activity but 5.8% ($p < 0.001$) lower amount of moderate activity (Fig. 3). Average time spent in sedentary or vigorous activity did not differ between ACC and EMG methods (Fig. 3). Bland–Altman plots further revealed heteroscedasticity in all but light PA categories so that the difference between the methods became accentuated with greater amount of time spent at the given category (Fig. 3).

## Comparison of different thresholds in ACC and EMG

Since the days with or without exercise had similar PA levels, the average of the days was used to compare different cut-off points in ACC and EMG estimated sedentary (/muscle inactivity), light, moderate and vigorous activity time. Figure 4 shows comparison of EMG and differently analyzed ACC results (with epoch of 15 s). ANOVA showed main effect of method ($p < 0.001$) and intensity ($p < 0.001$) with interaction ($p < 0.001$). When compared to EMG, Evenson's (*Evenson et al., 2008*) and Pate's (*Pate et al., 2006*) cut-off points yielded the smallest differences in sedentary time (less than 1% and 10%, respectively) among other methods (*Puyau et al., 2002*; *Sirard et al., 2005*; *Van Cauwenberghe et al.,*

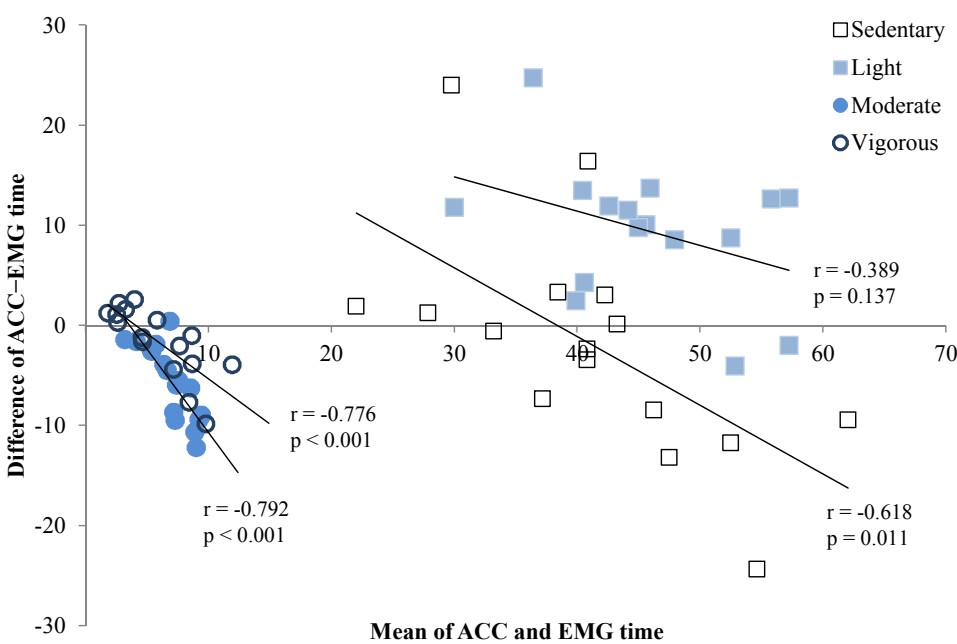

**Figure 3** **Bland–Altman plots showing differences in sedentary, light, moderate and vigorous activity time between ACC and EMG.** Positive values reflect ACC resulting in greater amount of time and negative values indicate EMG provides greater amount of time spent at given PA level. Significant linear correlations illustrate that heteroscedasticity was present at each category.

**Table 3** **Sedentary time and physical activity levels during days with and without exercise using accelerometry (ACC) and electromyography (EMG) methods.**

|  | Day with exercise | Day without exercise |
|---|---|---|
| ACC ($n = 13$) | | |
| Duration of recording (min) | $526.1 \pm 34.5$ | $521.9 \pm 38.9$ |
| Sedentary (%) | $38.7 \pm 7.1$ | $42.2 \pm 9.4$ |
| Light activity (%) | **$50.7 \pm 6.7$*** | **$48.3 \pm 7.8$*** |
| Moderate activity (%) | **$4.7 \pm 1.0$*** | **$4.5 \pm 1.9$*** |
| Vigorous activity (%) | $5.9 \pm 4.5$ | $5.0 \pm 2.4$ |
| EMG ($n = 7$) | | |
| Duration of recording (min) | $514.0 \pm 48.7$ | $513.5 \pm 47.1$ |
| Muscle inactivity (%) | $45.5 \pm 12.9$ | $41.4 \pm 16.8$ |
| Muscle light activity (%) | **$39.8 \pm 8.8$** | **$40.9 \pm 10.9$** |
| Muscle moderate activity (%) | **$9.3 \pm 3.1$** | **$10.0 \pm 4.6$** |
| Muscle vigorous activity (%) | $5.5 \pm 3.4$ | $7.7 \pm 7.3$ |

Notes.
*ACC significantly different from EMG method at given physical activity intensity, $p < 0.05$.

*2011*), which overemphasized sedentary time ($p \leq 0.001$). Furthermore, Evenson's and Pate's methods provided similar light and vigorous activity time (n.s.) as EMG. All other methods showed smaller light activity time ($p < 0.001$) compared to EMG. Moderate activity time was similar to EMG with Pate's methods, while with others being significantly

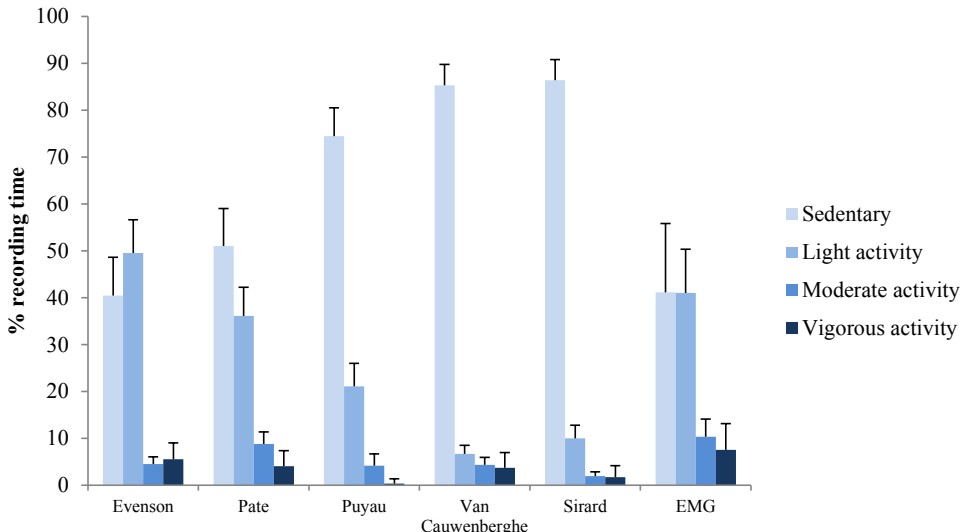

**Figure 4  Comparing physical activity intensities during children's day when analyzed using different cut-off points.** EMG is shown as a comparison. $n = 14$ (ACC), $n = 10$ (EMG).

different from EMG ($p \leq 0.008$). Vigorous activity using Puyau's ($p = 0.010$) and Sirard's methods ($p = 0.016$) was significantly smaller than using EMG, other methods did not show significant differences.

In the epoch-length analysis using Evenson's cut-off points and EMG we found main effect of epoch ($p < 0.001$) and intensity ($p < 0.001$) with interaction ($p < 0.001$). There was a progressive decrease in the time spent in sedentary (56% to 36%, $p < 0.001$) and increase in light-intensity activity (34% to 54%, $p < 0.001$) as the epoch length increased from 1 s to 30 s (Fig. 5). The time spent in moderate to vigorous intensity activity remained at about 10% of recording time between the 1–30 s epochs. With 60 s epoch the time spent in sedentary and light intensity activity dominated. Overall, the epochs of 7.5 s and 15 s showed the least difference with EMG method (less than 10% at each intensity level where the only significant differences were at moderate intensity, $p = 0.004$ and $p = 0.008$, respectively).

## DISCUSSION

The primary findings of this exploratory study were as follows. Firstly, in accordance with the hypothesis, ACC and EMG provided different results and interpretation of the intensity and duration of children's typical PA tasks. Muscle EMG activity was emphasised over ACC counts during crawling, static squat, single leg hops, standing long jump and jump for height, while ACC values were emphasised over EMG during swinging and trampoline jumping. During walking, swinging, trampoline jumping, stair negotiation and crawling ACC showed that over 60% of the time was spent in vigorous intensity while EMG allocated a high proportion of the time to light and moderate intensity. Secondly, during normal daily life ACC resulted in greater proportion of light activity but smaller proportion of

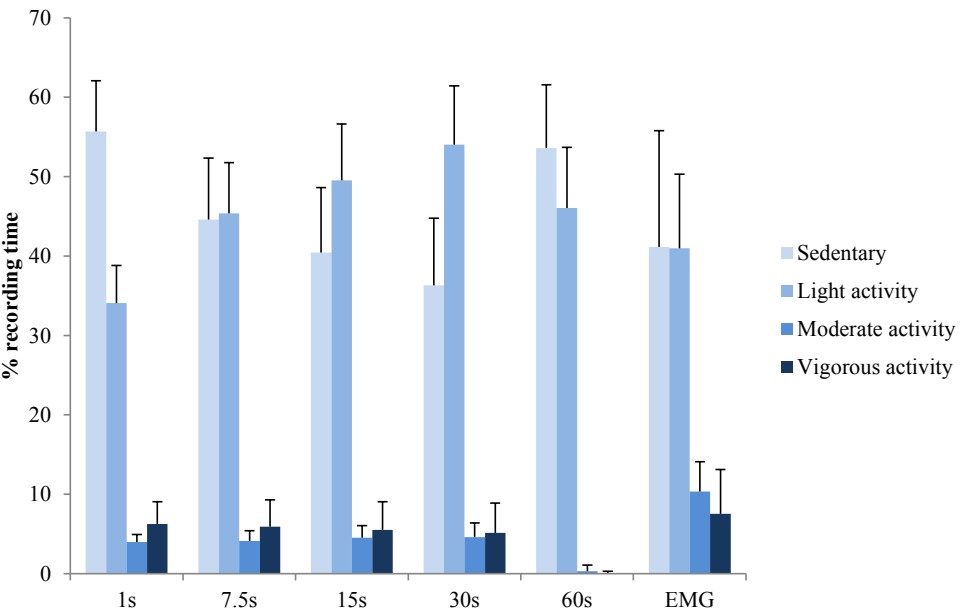

**Figure 5 The effect of different epoch durations on physical activity intensities and comparison to EMG assessed during children's daily lives with and without exercise.** The cut-off thresholds by Evenson's method were used. $n = 14$ (ACC), $n = 10$ (EMG).

moderate activity compared to EMG. Amongst the different thresholds (*Puyau et al., 2002*; *Sirard et al., 2005*; *Pate et al., 2006*; *Van Cauwenberghe et al., 2011*), the cut-off points by *Evenson et al. (2008)* with epochs of 7.5 s and 15 s yielded the smallest difference against EMG-derived PA levels.

Although ACC and EMG inherently measure different quantities, their assessments are interrelated. In the present study, we assessed a total of 13 different typical daily tasks with simultaneous measurements of ACC counts and EMG amplitude. When ACC and EMG values were both normalized to walking, the difference between ACC and EMG had the smallest difference during sitting, climbing, stair negotiation and game of tag (range of 14%–47%) as compared to the other tasks (Fig. 1). Specifically, in crawling, static squat, standing long jump, single leg hops and jump for height, EMG indicated over 170% higher relative intensity compared to ACC, while in swinging and trampoline jumping, the roles were reversed. This is reasonable since in trampoline jumping and swinging the body is hurled through the space resulting in high ACC counts, while only brief bursts of main locomotor muscle activity are observed in the EMG. On the other hand, in balancing and static squat, the body is quasi stationary while the main locomotor muscles are working to maintain the posture position. Predictably, this results in low ACC counts, while EMG activity is observed. The distribution of intensity levels in the typical PA tasks further highlights the methodological differences (Fig. 2).

In case of sporadic activities, EMG presumably results in a more realistic representation of energy demands while ACC counts summed over the duration dilute the effect. For example, during jumping tasks high EMG amplitudes in both push-off and landing phases

are necessary, and thus resulted in a great mean EMG. Contrarily, ACC counts were relatively low during standing long jump, single leg hops, and jump for height because high ACC peaks were only seen in during landing. Thus, it suggested that further insight could be achieved by comparing the counts with g-value analysis where raw ACC data provides more direct information of the impacts loading the body (*Laukkanen et al., 2013*; *Laukkanen et al., 2014*).

Increasing participation in organized PA has been suggested as a strategy for increasing overall PA in children (*Hebert et al., 2015*). However, in the present study with small sample, we did not find significant difference in either EMG or ACC PA levels between days with and without exercise (Table 3). It seems that one session of 60–90 min organized exercise may not alter the PA level of the whole day, although it provides a supportive environment for increasing PA during the organized session. Children may compensate their increased PA by relaxing outside the exercise session, and transportation to the exercise venue (especially during winter when some assessments were done) may require sedentary time that is not present on days without exercise (*Ridley, Zabeen & Lunnay, 2018*).

During the 9 h recordings, ACC and EMG provided slightly different values for the time spent at different PA intensities. While there were no significant differences between ACC estimated sedentary time and EMG estimated muscle inactivity time on either day with or without exercise, ACC yielded ∼9% more light PA time but ∼5% less moderate activity time than EMG (Table 3). Furthermore, the Bland-Altman plots showed heteroscedasticity for sedentary, moderate and vigorous activity time so that the difference between ACC and EMG became accentuated with greater amount of time spent at the given category (Fig. 3). Consequently, the choice of method influences interpretation; in the present sample, EMG allows the conclusion that the children meet PA recommendations having over 60 min MVPA while this was not reached according to ACC. Note, that the recordings were only 9 h and if the data was adjusted for 12 h, for example, also ACC data suggests that these children meet PA recommendations.

In this context, it is important to discuss the chosen EMG thresholds that were defined on individual basis while the ACC thresholds were the same for all individuals. Firstly, regarding EMG, the data requires normalization in order to compare amplitudes between individuals (*Vigotsky et al., 2018*). In this study, we normalized the EMG amplitude to that found during normal walking. Furthermore, the thresholds for moderate and vigorous PA levels were associated to individually chosen preferred walking speed of each child. While differences in the walking technique or speed may have influenced the EMG thresholds (*Lee & Hidler, 2008*), these differences do not pertain to ACC thresholds. In ACC count vs. EMG amplitude comparison both signals were normalized to walking, which enabled direct comparison in various PA tasks (Fig. 1). During walking, mean of 1218 ACC counts were recorded during walking, which is closer to moderate-to-vigorous ACC threshold than light-to-moderate ACC threshold and thus corresponds rather closely with the EMG threshold. On the other hand, the threshold for sedentary time was absolute EMG amplitude and during sitting it was 16% of walking. The corresponding mean ACC counts value was 18 during sitting that would have been categorized to light activity. This may be

due to the fact that the children were sitting on the floor instead of chairs in the current study. Furthermore, because this is the first study using EMG shorts in children, more validation work is required to inform of accurate, EMG-derived estimates of PA levels and their thresholds in children. In adults, the validation of muscle inactivity threshold is based on differentiating sitting from standing (*Pesola et al., 2014*; *Pesola et al., 2016*; *Gao et al., 2016*). In this study we aimed to differentiate muscle activity from inactivity, and set the threshold (3 μV) above signal baseline. This enabled recording of the smallest muscle activities also when sitting (*Pesola et al., 2016*), which is the strength of EMG. However, the definition and operationalization of muscle inactivity should be further validated in children.

It can be questioned whether comparison of methods that reflect different aspects of PA is reasonable in at the first place. However, PA is defined as body movement caused by muscle activity (*Caspersen, Powell & Christenson, 1985*), which supports the use of EMG to measure PA. Both metabolic and neuromuscular loading during PA are associated with specific benefits (*Strong et al., 2005*; *Keawutan et al., 2014*; *Poitras et al., 2016*). In many studies the PA exposure is measured with ACC, but the analyzed outcomes are more related to neuromuscular than metabolic loading of activity, e.g., motor skill competence (*Stodden et al., 2008*; *Lopes et al., 2012*; *Robinson et al., 2015*). Therefore, it is important to understand how ACC-derived PA relates to neuromuscular loading and the current study provides data to compare ACC data to directly measured muscle EMG activity.

Different cut-off points and epoch lengths were also compared in the present study. As expected, the choice of cut-off points and epoch lengths had significant effects on the time spent in sedentary, light, moderate and vigorous activities. For example, there was a progressive increase in the time spent in sedentary when ACC counts for sedentary is increased from 12 counts (*Evenson et al., 2008*) to 400 counts (*Sirard et al., 2005*). However, it should be noted that the published cut-off points were established in their own calibration study. It is recommended to follow the same criterion measures used in original study in order to make comparable and accurate results for the specific data sets (*Migueles et al., 2017*). The present study aimed to compare ACC with EMG-derived estimates of PA level. We found that Evenson's cut-off points (*Evenson et al., 2008*) yielded the smallest differences among other methods (*Puyau et al., 2002*; *Sirard et al., 2005*; *Pate et al., 2006*; *Van Cauwenberghe et al., 2011*) against EMG estimates. Since Evenson's cut-off points have been often used in children's studies (*Migueles et al., 2017*), it may serve as a surrogate method for EMG-derived estimates of muscle activity and inactivity time in young children.

Epoch lengths from 1 s to 60 s were explored using the Evenson's cut-off points (*Evenson et al., 2008*). We found a progressive decrease in the time spent in sedentary and increase in light-intensity as the epoch length was increased from 1 s to 30 s. The proportion of moderate- and vigorous-intensity activity were similar for the epoch lengths between 1 s and 30 s, but for the epoch length of 60 s the recording time involved mainly sedentary and light activity time. These observations may reflect children's typical sporadic activities and intermittent activity patterns during their daily life (*Fischer et al., 2012*; *Orme et al., 2014*), that are diluted with longer epochs. A previous study also compared different epoch lengths, and they found that using shorter epoch lengths (e.g., 15 s) compared with

60 s resulted in less sedentary time and higher MVPA time in children (*Ojiambo et al., 2011*). It seems that shorter epochs are recommended to capture short bouts of activity occurring frequently in young people (*Migueles et al., 2017*). *Bailey et al. (1995)* reported the majority of children's PA bouts occur in a short bouts. Furthermore, when compared with EMG-derived muscle activity levels, the epoch lengths of 7.5 s and 15 s had the smallest difference in the classification of PA levels, in which the range within 10% difference for each intensity level. Given that epoch length can influence the outcomes, it is important to use the same epoch length in order to compare different studies. Thus, based on the current findings, future studies may use the similar ACC epoch length of 7.5 s or 15 s for close association with muscle activity -based estimations of PA levels in children. However, it should be noted that the PA tasks analyzed had different durations. This was chosen so that in each task we could include sufficient number of repetitions (e.g., gait cycle, crawling cycle swing cycle, trampoline jump cycle) in order to provide accurate representation of the task in different PA intensities. Naturally the cycle time is shorter in gait compared to swinging thus influencing the chosen duration. On the other hand, some tasks are more difficult for children to sustain, such as the balancing task or maximal jump, where the time window for constant movement pattern is shorter. Consequently, we accepted variable durations of the analyzed tasks (Table 1).

Some limitations should be taken into consideration when interpreting the present results. First, the present research consists of two independent studies with a small sample size that may not be representative; age, motor coordination, body mass, lean body mass, fat mass, height or leg length, for example, may influence accelerations detected by the accelerometer and muscle activities assessed by EMG (*Sirard et al., 2005*; *Pesola et al., 2016*). Secondly, selection of the specific thresholds of EMG for muscle inactivity and activity levels affect the results because of the sensitivity of thresholds (*Klein et al., 2010*). When determining EMG thresholds we have relied on our experience with adults (*Tikkanen et al., 2013*; *Tikkanen et al., 2014*; *Pesola et al., 2016*) but in children, the EMG thresholds require a proper validation study with methodological comparisons, which is currently ongoing. Regarding methodological issues related to comparison of ACC and EMG signal processing, it is to be noted that attempts to process the raw signals in a similar manner should be made to allow more comprehensive methodological comparison and outcome characterization. Furthermore, we compared published cut-off points of ACC in our dataset, although our device and analysis process differed from the data collection procedures and processing criteria used in the original validation/calibration study. Overall, further studies should evaluate children's PA using various methodological approaches to gain understanding of not only the PA behavior but also of the significance of neuromuscular loading during development.

## CONCLUSIONS

By comparing accelerometer-derived and muscle EMG activity-based estimates of PA intensity and sedentary time this exploratory research provides novel insight into neuromuscular and metabolic loading of various PA tasks and daily life of school-aged

children. In particular, requirement of neuromuscular activity during PA tasks involving quasi static movements (balancing, crawling and static squat), sporadic activities (standing long jump, single leg hops, and jump for height) or movements associated with aerial maneuvers (trampoline jumping, swinging), seem to be poorly represented with ACC counts as compared to EMG. Exploration of different cut-off points and epoch lengths showed that Evenson's (*Evenson et al., 2008*) thresholds with ≤15 s epochs may best mimick EMG activity intensity categories, although during daily life ACC showed more time spent at light and less time at moderate intensity level than EMG. Since EMG and ACC inherently measure different aspects of children's PA, their combination can provide understanding on the quality of PA from both the neuromuscular (development of motor competence and coordination) and metabolic loading (overall movement and energy expenditure) perspectives.

### Funding
This study is funded by Ministry of Education and Culture (OKM/59/626/2016), Finland. The funders had no role in study design, data collection and analysis, decision to publish, or preparation of the manuscript.

### Grant Disclosures
The following grant information was disclosed by the authors:
Ministry of Education and Culture: OKM/59/626/2016.

### Competing Interests
The authors declare there are no competing interests.

### Author Contributions
- Ying Gao conceived and designed the experiments, analyzed the data, contributed reagents/materials/analysis tools, prepared figures and/or tables, authored or reviewed drafts of the paper, approved the final draft.
- Martti Melin, Arto J. Pesola, Arto Laukkanen and Taija Finni conceived and designed the experiments, performed the experiments, analyzed the data, contributed reagents/materials/analysis tools, authored or reviewed drafts of the paper, approved the final draft.
- Karoliina Mäkäräinen conceived and designed the experiments, performed the experiments, analyzed the data, approved the final draft.
- Timo Rantalainen analyzed the data, contributed reagents/materials/analysis tools, authored or reviewed drafts of the paper, approved the final draft.
- Arja Sääkslahti conceived and designed the experiments, analyzed the data, contributed reagents/materials/analysis tools, authored or reviewed drafts of the paper, approved the final draft.

## Human Ethics

The following information was supplied relating to ethical approvals (i.e., approving body and any reference numbers):

Both Study I (26.8.2014) and Study II (25.8.2012) received ethics approval from the Ethics Committee of the University of Jyväskylä.

## Data Availability

The raw data are provided in Supplemental Information 1.

## Supplemental Information

Supplemental information for this article can be found online at http://dx.doi.org/10.7717/peerj.5437#supplemental-information.

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
