# Peer review of "Children’s physical activity and sedentary time compared using assessments of accelerometry counts and muscle activity level"

_PeerJ, doi:10.7717/peerj.5437_

## Round 0.1 · original submission · Major Revisions

Dear authors,

Your manuscript has been carefully reviewed by three experts, which have indicated scientific merit in your work. However, they have suggested some changes which should be taken into account before considering your work for publication (MAJOR REVISION).

With respect and warm regards,
Dr Palazón-Bru (academic editor for PeerJ)

·

Basic reporting

The English used throughout the manuscript is professional and generally clear and concise. However, some sentences contain grammatical errors (examples: line 61, 87) or are slightly long and sometimes hard to read (example: line 87-92). The manuscript would benefit from another proofread.

The background provided is sufficiently backed up with quality references and background and context is provided. Please add references to line 61 and 101. Also, the reference used in line 120 is inaccurately used; the references used only back up the first part of the sentence. Placing these references at the end of the sentence seems to suggest that previous research supports entire statement. Also, in line 244 is an adult reference provided whilst participants in the study are children; this should be noted.

The article has a professional structure and the figures and tables are clear. Raw data is shared and is clear.

Experimental design

The research falls under the aims and scope of the journal. The research question is well defined, relevant and meaningful.

The research is novel, interesting, and meaningful. Currently, accelerometry is the most commonly used measure of PA and SB, however it has it’s limitations. Therefore, it is important to compare this measure with other measures of PA and SB. The research conducted contributes to the knowledge around measure PA and SB. This has been clearly explained in the manuscript.

The methods used in the research were limited with regards to the following issues;
1) Number of participants: Whilst 2 samples of participants were combined within this study (n = 11 and n = 14), samples were only combined in the "walking" task. The remaining tasks were different across the 2 studies and only assessed for either 11 or 14 participants. For example, the results in the sections ‘PA levels in specific tasks’, ‘PA levels during daily life’ and ‘comparison of different thresholds in ACC and EMG’ were therefore only based on 14 participants. Interestingly, the differences between ACC and EMG for “walking” was the smallest of all tasks (figure 1) – this is the only activity assessed in all 25 participants. The low number of participants should be noted as a limitation.

2) Duration of tasks/Activities assessed: The activities assessed are very short in duration (all under 2 minutes). For ACC measures, data were summed over 15 second epochs (for all analyses other than the epoch length comparisons). In other words, the assessed tasks ranged in duration from 1 to 8 epochs. Please explain to what extend the provided %'s spent in certain intensities (presented in figure 2, for ACC) are representative for the actual % spent in intensities, or that combination of the short duration of tasks with relatively long epoch times might have limited the variation in %'s within the tasks. Interestingly, ACC values were emphasized over EMG values in the longer activities (swinging and trampoline jumping, line 327), and EMG values were emphasized over ACC values in shorter activities (crawling, squat, etc., line 325). Also, it was not fully clear to me whether the EMG data had been collected in shorter frequencies or with similar "epoch lengths". Did the researchers consider longer activities or analyses with event files rather than epochs? This should be stated as a limitation.

3) The lack of validated thresholds for EMG in children. The EMG shorts have not been validated in children. The authors recommend to use the ACC epoch length of 7.5s or 15s for close association with muscle activity based estimations of PA in children, however, given the methods to measure EMG in children have not been validated this might be premature. This has been mentioned in the discussion, but this could be extended.

The methods have been described in sufficient detail and could be replicated. Personally, I think the provided detail on the tasks (list of activities) is slightly too long and would fit better in an attachment. Most of the activities are easy to understand from the name itself and the extensive explanation is not needed within the manuscript.

Validity of the findings

Whilst I believe the research is novel and important, I think the recommendations made and conclusions drawn are too strong for the provided evidence. For example, in lines 418-419 and in lines 435-437. Given the previously mentioned methodology issues, I think it’s too premature to make these strong conclusions.

Hence, I think the recommendations and conclusions should be softened and more attention should be given to the limitations in the study design. I think the study could inform future studies on how to collect more evidence using similar approaches, rather than drawing conclusions and making recommendations based on just the results provided in the manuscript.

Additional comments

This study compared accelerometer (ACC) -derived and muscle electromyography (EMG) -based estimates of physical activity (PA) and sedentary time in common PA tasks and during daily life of children. The study combined data from 2 separate studies, N = 25 in total.

This paper is addressing an important research gap. Currently, accelerometry is the most commonly used measure of PA and SB, however it has it’s limitations. Therefore, it is important to compare this measure with other measures of PA and SB. The research conducted could contribute to the knowledge around measure PA and SB.

Whilst I think the findings of this study are important and novel, I think the recommendations made are premature. The participant number is low (especially because the participants of 2 studies are only combined in “walking”), the activities assessed were of short duration (i.e., containing only a few epochs) and EMG shorts thresholds have not been validated in children yet. Hence, I think the recommendations and conclusions should be softened and more attention should be given to the limitations in the study design. Given the novel techniques, I think the study could inform future studies on how to collect more evidence (e.g., bigger sample size) using similar methods. Alternatively, the current study could increase participant numbers and include longer activities.

Line: 67-68: Are there any systematic reviews which you could add here as a reference?
Line 79-80: Please explain this a little more, especially for researches who have not used EMG before.
Line 96-100: Very important information, I personally would prefer to read this earlier in the discussion.

Recruitment and study sample: It was not clear to me how many participants dropped out due to unsuccessful ACC and/or EMG recordings in study 2 (i.e., missing data). Did study 2 actually include data from 14 participants or were the analyses only done for the participants who did not miss data (only 7 participants, based on line 146?)?

Line 162: Why did the authors use waist-accelerometers and not hip-accelerometers?

Table 1: Are these numbers means? Please provide this information.

Line 226: Why were the Freedson cutpoints not included in these analyses?

Line 242-243: Please add information regarding the validation here. I believe these references only included adult research. Alternatively, report that this is information based on adult research.

Discussion: The discussion is very long and could be shortened, for example by:
Line 334-341: This could be added to the introduction rather than the discussion.
Line 363-371: This is not very relevant to the current study and could potentially be removed from the manuscript.

Reviewer 2 ·

Basic reporting

I found the standard of basic reporting to be good, with a sensible article structure. I could logically follow, from the raw data, what the authors did.

I believe the tables need to be reformatted to a higher standard than currently present. I also believe the table captions could be more elucidatory.

the results logically emerge from the methods, and methods from the aims/hypothesis.

I felt that the standard of English needs to be sharpened up throughout the manuscript. The introduction, in particular, I noted a number of occasions sentences were unclear, should be reworded or greater clarity is required. For instance; first line "a child has an inborn drive to be physically active" - very much a throw away sentence and should be cited and subsumed into the following sentence.
line 64 - examples of chronic diseases,
line 68 - "sedentary behaviour increases with age", you need to elaborate on this point, by what magnitude or we talking? I feel this is an important point given the importance of sedentary time/behaviour to this manuscript.
line 71 - specify what ages you mean with "school aged and youth".
line 72-76 - this read overly verbose, and could be much more concise
line 77 - reference use of accelerometry "widely".
line 77 - 80 - please be careful with a very blanket description of accelerometry, there are many types of accelerometer available that work quite differently, including; MEMS, piezoelectric, piezo-resistive, capacitative.
line 83 - better description/definition of METS
line 86 - 88 - English standard is poor here
line 84 - by "observation", do you mean through tools like SO-PLAY? or just that MET values are apparent? please clarify
line 92 - 95 - please mention band-pass filters (high and low) also, these can have just as serious impact on accelerometry.

Within the abstract I would recommend you focus the discussion section on what "new" findings this work presents. Of particular interest is neuromuscular loading component. whereas referral to the impact of various cut-points is well-established.

Experimental design

the study is well within the scope of the journal, with a reasonably well defined.

I need the authors to reassure/explain to me (and the prospective reader) the rigour involved in combining 2 separate studies, with small sample sizes, looking at different activities and different ages.

furthermore, throughout the methods section (this may, in part, help address the previous comment), far more, explicit, detail is required. I am particularly au fait with accelerometry and sensors, but I struggled to follow precisely what happened methodologically. Please clarify in precise detail here. I think it is doubly important given the combination of 2 studies, and in order to facilitate replication, I recommend a very tight re-write of this section to remove all elements of ambiguity.
also curious why the use of Bonferroni? this is a very conservative?

Validity of the findings

I found the results to be logically inferred from the data in most cases.

However, in the PA levels during daily life, the initial omnibus test (ANOVA) failed (non-significant), yet you continued to run a post-hoc test, I do not believe this is valid.

If you commence your investigation with the ANOVA and it fails, you have no license to continue investigating.

I think if you want to refer to the differences between the days, perhaps consider the use of effect sizes.

the rest of the results I was happy with.

Additional comments

Overall, I felt this meets all of the requirements and scope of the journal, with primarily only minor re-writing/editing in the manuscript. My only substantive concern was the methodological rigour, and clarity of the methods, which I would advise the authors address.

Reviewer 3 ·

Basic reporting

This is an interesting paper that compares estimates of physical activity obtained from a hip-worn accelerometer with measures of muscular activity measured by EMG. This is done by comparing the measured intensity of various activities (using % of walking intensity, and % of time spent within each intensity category). There is also a comparison of different activity count intensity thresholds (cut-points) and different epoch lengths. The main conclusion from this work is that cut point/epoch decisions can affect results, and using the Evenson (2008) cut points with an epoch <15 s best represents the EMG data.

Although intelligible, some parts of the paper are hard to read, and there are numerous small spelling/grammar errors. For a reader to get the most out of the paper, it should be easy to read. I feel it needs to be proofread by a colleague who is proficient in English. A few examples:

L31: and game of tag -> and a game of tag
L61 Physically active -> A physically active
L65 recommendations -> recommendations
L72-6 Very long sentence
L83 against derived from (doesn’t make sense)
L88 has been reported to be used -> have been reported and used
L87-92 Very long sentence
L92 Besides of the cut-off -> Besides the cut-off
L139 volunteered to -> volunteered to participate in
L140 were finally included in the final sample -> were included in the final sample
L430 reference error
Etc.

Experimental design

In general, the study is interesting, but the rationale for undertaking such work isn’t well articulated. I understand the purpose of the study is to compare the X6-1a accelerometer with EMG, but why this is important, the knowledge gap it fills, and how it will be useful is not well defined. In the introduction, the problems associated with count-based cut points are highlighted, particularly the inability to detect activity type and posture (i.e., standing vs sitting). These issues have been well publicised over the last 10 or so years (which is why much recent work has been moving away from this approach altogether). It is true that EMG provides a direct measure of the electrical activity of the muscle, and may thus better represent low intensity and sporadic activities. However, these two techniques are fundamentally different. Is this study attempting to validate the X6-1a accelerometer using EMG as the criterion measure? Or simply highlighting issues with cut points? Or is it proposing that EMG is an alternate/better/different measure than accelerometers? This should be clarified.

A few small comments about the methods:

Only the Evenson cut points were tested at different epoch lengths. What was the rationale for choosing these and not the other cut points?

Why was walking chosen as the activity upon which to normalise the data? Would sitting be more intuitive?

As pointed out in the introduction, the distinction between sitting and quiet standing is important. Is there a reason that quiet standing was not included in the activity protocol?

I wonder if a measure of agreement (e.g., kappa coefficient) would be useful for interpreting how these intensity categories align?

Validity of the findings

A potential problem with the design of the study is that you are using an accelerometer that may not be compatible with the cut point thresholds you are using. Most of these cut points were developed for ActiGraph counts (which are derived using a proprietary algorithm), using the vertical (y) axis only. I haven’t seen the X6-1a accelerometer in the literature before. Your treatment of raw accelerometer data to derive counts is not fully explained, and this sensor uses a different sample rate and different filtering techniques compared to the ActiGraph. I see you have mentioned this in the study limitations, but the validity of all the results you have presented are based on how closely the X6-1a counts represent the ActiGraph counts. Do you know of any previously published work that can demonstrate their comparability?

It is unclear how the EMG data were categorised into four intensity categories. From L248-252: < 3 μV is sedentary, mean during walking is light/moderate, and this doubled is moderate/vigorous. This is three categories, how did you separate light from moderate, and moderate from vigorous? Did you measure EMG during standing and take 90% as sedentary? Similarly, almost half of the data in study 1 and 2 was lost due to insufficient EMG signals. What is the validity/reliability of the EMG shorts? Particularly in 6-9-year-old children. This may seem trivial, but all the comparisons you make in this study are dependent upon the accuracy of these EMG data processing decisions.

Additional comments

To summarise: a thought-provoking paper that is a little unclear in what it set out to achieve. It needs clarification in several areas (particularly the validity of methods) before it will serve as a meaningful publication for others.

---

## Round 0.2 · Major Revisions

Dear authors,

Your manuscript has been re-reviewed by two of our expert referees, who found your paper to be of interest. However, there were some points at issue, as raised by Reviewer 1, that deserve your attention and major revisions must be made according to their comments.

With respect and warm regards,
Dr Palazón-Bru (academic editor for PeerJ)

·

Basic reporting

In general the English used throughout is professional.
Whilst the authors mention that a proofread has taken place, the manuscript still contains grammatical errors and long sentences. For example:
Line 45-47: "During day with and without exercise ACC resulted in greater proportion of light activity (p < 0.01) but smaller proportion of moderate activity compared to EMG (p < 0.05)."
Line 104-107: "It is important to note that during early years not only the cardiorespiratory (metabolic) but neuromuscular system in particular plays an important role because a major contributor to the development of motor performance, fitness, as well as proficiency of gross motor skills, is the ongoing neuromuscular development (Haywood & Getchell, 2014)."
Please ensure that the manuscript gets another proof read to address the missed errors.
The background provided is sufficiently backed up with quality references and background and context is provided. The article has a professional structure and the figures and tables are clear. Raw data is shared and is clear.
FinalComments_response_v2.docx: I would like to recommend the authors to provide some more guidance through the response document, for example through page and line numbers referring to the changes made. It was at times hard to find the addressed comments in the actual manuscript and I might have potentially missed valuable changes.

Experimental design

The research falls under the aims and scope of the journal. The research question is well defined, relevant and meaningful.
In my previous review, I expressed some concern regarding the duration of tasks/activities assessed in the previous manuscript. It is now clear to me why the choices were made and how the duration was influenced my certain factors. Thanks for clarifying this.
In general, I do think that the authors have provided a more comprehensive explanation about the methods and experimental design and this section has much improved. However, I still have some issues with the manuscript, which I have mentioned below.
Whilst the authors mention that they are aware of the small sample size, they only added limited text about this in the discussion (a small change in line 516). Some more specific limitations and potential implications for future research with a higher sample size could be added. The authors mention that an ongoing study will include more children, at the moment I do not understand in what areas that study will be improved/innovative compared to the current study and why the choice was not made to pool this into one study. It might be a good addition to further expand what future studies should incorporate. Did the authors contemplate classifying this study as a pilot study?
In the previous review, I asked for some clarification about epoch lengths/sampling rates:
- Comment: Also, it was not fully clear to me whether the EMG data had been collected in shorter frequencies or with similar "epoch lengths". Did the researchers consider longer activities or analyses with event files rather than epochs? This should be stated as a limitation.
The authors answered the comment as follows:
- [R]: EMG and ACC were always analyzed over the same duration of the task (table 1). In this manuscript we have used the traditional counts-based analysis for ACC. Although ACC is collected with higher sampling rate, the data is reduced by using certain epochs. In future it would be good to use methods allowing raw data analysis (e.g. Vähä-Ypyä et al. 2015). EMG, is also typically analyzed in some epochs, e.g. average rectified value of 1 s is typically used for maximal voluntary contraction. In the current methodology, the available minimum signal is average rectified value every 40 ms. By taking a mean value of this for the duration of each task is not different if we had first averaged it in 1s windows and then during the entire duration of the task. Thus, while we may perceive the “effective epoch” length of ACC and EMG being different, by changing EMG processing we do not gain differences but ACC should be completely analyzed using different principles. In the current manuscript we chose to report ACC in counts that is the current standard.
In addition, this was stated as responses part of another reviewer’s comments:
- For example, we found some sporadic activities (standing long jump, single leg hops, and jump for height), which EMG presumably results in a more realistic representation of energy demands while ACC counts summed over the duration dilute the effect (although ACC data processing issues are interrelated to this problem but not dealt with comprehensively in the manuscript).
Based on the previous comments and responses, it is still not clear as whether the choice of ACC counts in combination with EMG was compatible. Whilst I acknowledge that my lack of knowledge about EMG might be partially why this is still unclear, I do believe that this should be further explained to the reader also.

Whilst the authors have acknowledge in their manuscript that validation work is currently in progress, this does not cover the current research and is obviously depending on the results of this study. Are any unpublished results available which could serve as a justification for the chosen methods?

Validity of the findings

The recommendations and conclusions are slightly softened and more attention should is given to the limitations in the study design. Given the issues in the methodology, I still think that the results and conclusions are quiet strong. I think the paper could serve more as exploratory work informing future studies.
One minor comment regarding the following response:
- [R]: We have been modified texts (original lines 418-419 and lines 435-437) accordingly.
I was initially unable to find these lines in the current manuscript. I recommend to provide line numbers of the newest manuscript rather than the original manuscript.

Additional comments

Most comments have been addressed appropriately and the explanation was clear, thank-you.
Nevertheless, there were still some grammatical errors throughout the manuscript. My biggest concern is the fact that the EMG method used is not validated in children yet. Whilst it is great to see that the authors are currently working on the validation, I think it would be really valuable to gain research of that research and implement this into the current paper. That would strengthen the current paper a lot and justify the drawn conclusions. Given the issues in the methodology, I still think that the results and conclusions are quiet strong. I think however that the paper could serve as exploratory work informing future studies.

Reviewer 2 ·

Basic reporting

Thank you, to the authors, for diligently and forthrightly answering my own and the other reviewers comments. I found your rebuttal to logical and agreeable. As such, I found the basic reporting to be of a good standard.

Experimental design

The authors did a great job here of refuting some of the edits/comments put forward by the reviewers. In doing so, you have assuaged any doubts in my own mind, and am more the happy with this section.

Validity of the findings

all fine here. The authors, again, respectfully and soundly rebutted all comments.

Additional comments

Thank you for carefully attending to all of my own and other reviewers comments. I hope you feel/agree that the comments were fair and warranted. I also hope you feel that they have helped improve the manuscript. Following your diligent and soundly written responses, I am more than happy to suggest acceptance for publication of this novel work. Well-done, and I can't wait to see what more comes from this line of inquiry.

Dr Cain Clark

---

## Round 0.3 · accepted · Accept

Dear authors,

All the comments of the reviewers have been correctly addressed in this new version of the manuscript. Therefore, I am happy to inform you that your paper has been accepted for publication in PeerJ.

Congratulations!

With respect and warm regards,
Dr Palazón-Bru (academic editor for PeerJ)

·

Basic reporting

In general the English used throughout is professional.
Sufficient references and background provided.
The article is structured professionally.

Experimental design

The research falls under the aims and scope of the journal. The research question is well defined, relevant and meaningful.

Validity of the findings

The authors have added more emphasis on that this is an exploratory study, conclusions are softened, and more attention is given to the limitations in the study design.

Additional comments

I am satisfied with the authors' responses.